# Epidemiological and genomic investigation of chikungunya virus in Rio de Janeiro state, Brazil, between 2015 and 2018

Filipe Romero Rebello Moreira[1,2], Mariane Talon de Menezes[1], Clarisse Salgado-Benvindo[1,3], Charles Whittaker[2], Victoria Cox[2], Nilani Chandradeva[2], Hury Hellen Souza de Paula[4], André Frederico Martins[4], Raphael Rangel das Chagas[4], Rodrigo Decembrino Vargas Brasil[4], Darlan da Silva Cândido[2,5], Alice Laschuk Herlinger[1], Marisa de Oliveira Ribeiro[6], Monica Barcellos Arruda[6], Patricia Alvarez[6], Marcelo Calado de Paula Tôrres[1], Ilaria Dorigatti[2], Oliver Brady[7,8], Carolina Moreira Voloch[1], Amilcar Tanuri[1], Felipe Iani[9], William Marciel de Souza[10,11], Sergian Vianna Cardozo[4], Nuno Rodrigues Faria[2,5,12]*, Renato Santana Aguiar[13,14]*

1 Departamento de Genética, Universidade Federal do Rio de Janeiro, Rio de Janeiro, Rio de Janeiro, Brazil, 2 MRC Centre for Global Infectious Disease Analysis, Jameel Institute, Imperial College London, London, United Kingdom, 3 Department of Medical Microbiology, Leiden University Medical Center, Leiden, The Netherlands, 4 Departamento de Saúde, Programa de Pós-graduação em Biomedicina Translacional, Universidade do Grande Rio (UNIGRANRIO), Duque de Caxias, Rio de Janeiro, Brazil, 5 Department of Zoology, University of Oxford, Oxford, United Kingdom, 6 Institute of Technology in Immunobiology Bio-Manguinhos, Oswaldo Cruz Foundation/ Fiocruz, Rio de Janeiro, Brazil, 7 Centre for the Mathematical Modelling of Infectious Diseases, London School of Hygiene & Tropical Medicine, London, United Kingdom, 8 Department of Infectious Disease Epidemiology, Faculty of Epidemiology and Population Health, London School of Hygiene & Tropical Medicine, London, United Kingdom, 9 Fundação Ezequiel Dias (FUNED), Belo Horizonte, Minas Gerais, Brazil, 10 Department of Microbiology and Immunology, University of Texas Medical Branch, Galveston, Texas, United States of America, 11 World Reference Center for Emerging Viruses and Arboviruses, University of Texas Medical Branch, Galveston, Texas, United States of America, 12 Instituto de Medicina Tropical, Faculdade de Medicina da Universidade de São Paulo, São Paulo, São Paulo, Brazil, 13 Departamento de Genética, Ecologia e Evolução, Universidade Federal de Minas Gerais, Belo Horizonte, Minas Gerais, Brazil, 14 Instituto D'or, Rio de Janeiro, Rio de Janeiro, Brazil

* n.faria@imperial.ac.uk (NRF); santanarnt@gmail.com (RSA)

**Data Availability Statement:** The novel viral genome sequences were deposited to NCBI GenBank under Accession Numbers OP312936 to

## Abstract

Since 2014, Brazil has experienced an unprecedented epidemic caused by chikungunya virus (CHIKV), with several waves of East-Central-South-African (ECSA) lineage transmission reported across the country. In 2018, Rio de Janeiro state, the third most populous state in Brazil, reported 41% of all chikungunya cases in the country. Here we use evolutionary and epidemiological analysis to estimate the timescale of CHIKV-ECSA-American lineage and its epidemiological patterns in Rio de Janeiro. We show that the CHIKV-ECSA outbreak in Rio de Janeiro derived from two distinct clades introduced from the Northeast region in mid-2015 (clade RJ1, $n = 63/67$ genomes from Rio de Janeiro) and mid-2017 (clade RJ2, $n = 4/67$). We detected evidence for positive selection in non-structural proteins linked with viral replication in the RJ1 clade (clade-defining: nsP4-A481D) and the RJ2 clade (nsP1-D531G). Finally, we estimate the CHIKV-ECSA's basic reproduction number ($R_0$) to be between 1.2 to 1.6 and show that its instantaneous reproduction number ($R_t$) displays a strong seasonal pattern with peaks in transmission coinciding with periods of high *Aedes aegypti* transmission potential. Our results highlight the need for continued genomic and

OP312969. Data and code used in this study were made available in supplementary files and on the project GitHub repository at https://github.com/filiperomero2/CHIKV_RJ_2015-2018 and https://github.com/filiperomero2/ViralUnity.

**Funding:** This research was funded by the Rede Corona-ômica BR MCTI/FINEP affiliated to 116 RedeVírus/MCTI (FINEP 01.20.0029.000462/20, CNPq 404096/2020-4); MEC/CAPES 118 (14/2020 - 23072.211119/2020-10), FINEP (0494/20 01.20.0026.00), UFMG-NB3, FINEP n˚ 1139/20 (RSA) and FAPERJ (R.S.A 202.922/2018). This project was also partially funded by the Instituto Todos pela Saúde-ITpS (Chamada 01/2021-C1294). CW is supported by Sir Henry Wellcome Postdoctoral Fellowship (Ref 224190/Z/21/Z). The funders had no role in study design, data collection and analysis, decision to publish, or preparation of the manuscript.

**Competing interests:** The authors have declared that no competing interests exist.

epidemiological surveillance of CHIKV in Brazil, particularly during periods of high ecological suitability, and show that selective pressures underline the emergence and evolution of the large urban CHIKV-ECSA outbreak in Rio de Janeiro.

## Author summary

Chikungunya is a mosquito-borne viral disease that has emerged as a significant public health concern in many regions worldwide. The state of Rio de Janeiro, a key economic and tourism hub in Brazil, has experienced multiple outbreaks of chikungunya in recent years, resulting in significant morbidity and economic burden. However, our understanding of the establishment and epidemiology of chikungunya in Rio de Janeiro remains limited. We conducted an analysis of chikungunya epidemiology in Rio de Janeiro, focusing on the first four years of virus circulation in the state. We estimated the magnitude of chikungunya transmission during this period, providing evidence that the trends of incidence and transmissibility can be influenced by climatic fluctuations, which impact vector abundance. Through analysis of novel genetic data, we inferred the evolutionary history of the virus, providing evidence of molecular adaptation in the lineage circulating in Rio de Janeiro. These findings highlight the need for designing effective vector control strategies to prevent and mitigate chikungunya in Rio de Janeiro and similar settings. Further research is needed to continue monitoring viral epidemiological trends and to better understand the complex interactions between climatic factors, vector dynamics, and viral evolution in shaping the transmission of this emerging disease.

## 1. Introduction

Chikungunya virus (CHIKV) is an enveloped virus with a single-stranded positive-sense RNA 11.8 kb genome that belongs to the *Alphavirus* genus in the *Togaviridae* family [1]. The CHIKV genome contains two open reading frames encoding four non-structural proteins (nsP1–nsP4), important for viral replication, and four structural proteins (capsid, E1–E3), necessary for virus assembly [2]. CHIKV is transmitted by the bites of infected *Aedes* spp. mosquitoes, mainly from the *Ae. aegypti* species and, to a lesser extent, the *Ae. albopictus* species. In humans, clinical manifestations of chikungunya include arthralgia, high fever, myalgia, headache and often exanthema. However, chikungunya may also cause long-lasting debilitating polyarthralgia [1], and can also lead to neurological complications and fatal outcomes [3]. Despite being an important public health threat with >1 billion people at risk for transmission [4], reporting of chikungunya infections often relies on syndromic surveillance, which is challenged by the co-circulation of mosquito-borne viruses that cause similar clinical symptoms, including dengue (DENV) and Zika (ZIKV) [5].

Since its first description in Tanzania in 1952, chikungunya has caused over 70 outbreaks in Africa, Asia, Americas, Europe and in Ocean Pacific Islands [4]. CHIKV can be classified into four distinct lineages: the West-African lineage, the Asian lineage, the East-Central-South-African (ECSA) genotype [6], and the Indian Ocean lineage (IOL). Chikungunya's geographic distribution–particularly of Asian, IOL and ECSA lineages–has been expanding rapidly over the last 20 years probably due rapid urbanization, globalization of trade, and virus evolution and adaptation to local variation in the distribution of vector species [7–10]. For example, a single amino acid change in the E1 protein (*i.e.*, E1-A226V) of the CHIKV IOL lineage has been linked to increase transmissibility and infectivity in *Ae. albopictus* and was linked with a series of severe outbreaks in Indo-Pacific Asia [11–13].

In 2013, the CHIKV Asian lineage was first detected in St Martin islands in the Caribbean [14]. Since then, this lineage has spread to >50 countries and territories in the Americas [14,15]. In 2014, co-circulation of the Asian and ECSA lineages was detected in Brazil [16]. Local transmission of the Asian genotype was first detected in Amapá state, North Brazil, on 31 July 2014 [16]. On 25 August 2014, local transmission of the CHIKV ECSA lineage was identified in Bahia state, Northeast Brazil [16,17]. Retrospective outbreak investigations revealed that the ECSA index case in Brazil arrived in Bahia in late May 2014 from Angola [17]. This new ECSA-American lineage was then linked to large outbreaks in the Amazonas and Roraima states in North Brazil, despite earlier circulation of the Asian lineage in that region [18]. Currently, circulation of ECSA-American lineage has been confirmed in all five geographic regions of Brazil [18–23], but also in Paraguay [24] and Haiti [25].

Rio de Janeiro (RJ) is the third most populous Brazilian state and a key business and touristic hub in the Americas. Due to air and fluvial connectivity and high mosquito climatic suitability (adequate levels of temperature, humidity and precipitation for mosquito occurrence) [26], the state has historically suffered large mosquito-borne viral outbreaks [27], including dengue, Zika, yellow fever and, more recently, chikungunya. Between 2013 and 2018, CHIKV has caused 403,828 confirmed infections in Brazil [28]. In 2018, RJ became the epidemic center of CHIKV in Brazil, accounting for 41% of the cases reported in the country [28]. Until 2017, only 20,251 cases had been confirmed in Rio de Janeiro, corresponding to 6% of the total case numbers reported in Brazil. The earliest autochthonous chikungunya cases were reported in November 2015, and since then two ECSA sub-lineages have been detected in Rio de Janeiro [20,21,29]. Despite the burden imposed in Rio de Janeiro, chikungunya's evolution and transmission patterns remain poorly understood. The reliance of chikungunya diagnosis on clinical-epidemiological criteria and the scarcity of chikungunya sequence data from Rio de Janeiro, hamper our understanding of chikungunya transmission and evolution. Moreover, relevant epidemiological parameters, such as basic ($R_0$) and time varying ($R_t$) reproduction numbers, which measure virus transmissibility, have not been yet previously estimated for CHIKV in Rio de Janeiro, hindering comparisons with the virus' epidemiological dynamics in other settings.

Using genetic analysis of CHIKV newly generated and publicly available genome sequences and CHIKV traditional surveillance data, we investigated its evolution and transmission dynamics in Rio de Janeiro. We show that the large 2018 CHIKV outbreak in Rio de Janeiro was mainly caused by one dominant ECSA-American viral clade that likely persisted locally since mid-2015. In addition, we provide evidence of adaptive molecular evolution on non-structural amino acid positions, including in a lineage-defining mutation related with the dominant ECSA viral clade associated with the 2018 outbreak in Rio de Janeiro. Finally, using epidemiological modeling, we estimate basic and instantaneous reproduction numbers for CHIKV ECSA-American in Brazil, further expanding our understanding of the epidemiology of this rapidly expanding lineage.

## 2. Materials and methods

### 2.1. Ethics statement

Clinical samples were collected with formal written consent following approval of the ethical review board of *Universidade do Grande Rio* (approval number: 54544316.3.0000.5283).

### 2.2. Quantifying CHIKV transmissibility in Rio de Janeiro

The weekly aggregated epidemiological data of chikungunya laboratory-confirmed cases in RJ was obtained from the Brazilian Ministry of Health from 2014 to 2018, and then the results

were presented in time series. Data was used to estimate weekly CHIKV incidence and transmissibility (reproduction number) for Rio de Janeiro between 2016 and 2018. We estimated the basic reproductive number ($R_0$), defined as the number of secondary infections caused by an infected individual in a completely susceptible population, using data from 3 January to 13 March of 2016. We employed the exponential growth method to estimate $R_0$ [27], which derives an estimate of $R_0$ from the generation time distribution along with an estimate of the exponential growth rate, calculated during the earliest weeks of the outbreak where epidemic growth was approximately exponential and factors such as susceptible depletion, behavioral change or vector control are negligible. To assess changes in transmission over time in subsequent epidemics when immunity due to prior natural infection is present, we also calculated the instantaneous reproduction number ($R_t$), defined as the average number of secondary cases caused by an infected individual during the epidemic, provided the conditions remained as they were at time $t$ [30]. We used an 8-week sliding-window to estimate $R_t$, and conducted several sensitivity analyses considering different sliding-window intervals (3 to 7 weeks). We assumed a parametric gamma distribution for the generation time (GT) with a mean of 14 days and a standard deviation of 6.4 days, based on an earlier estimate obtained during an outbreak caused by the Indian Ocean lineage [31]. We also performed sensitivity analyses with different mean GT values (10 and 20 days), keeping the standard deviation constant. All analyses were performed in R software v4.2.2 [32] with the $R_0$ [33] and EpiEstim [30] packages.

## 2.3. *Aedes aegypti* transmission potential in Rio de Janeiro

To estimate *Aedes aegypti* transmission potential for RJ between 2014 and 2018, we first extracted daily values (midday) of 2m air temperature, 2m dew point temperature and total precipitation for all locations across RJ from the ERA-5 ECMWF reanalysis product [34]. We calculated relative humidity based on daily temperature and dew point estimates using the August-Roche-Magnus approximation [35–37]. Each variable was then averaged over RJ and combined to calculate index P using the MVSE R package [38]. We used priors related to *Ae. aegypti* ecology and entomology, as described elsewhere [38]. We then aggregated resulting daily estimates to generate mean weekly estimates for index P between 2014 and 2018. Code and data used in these analyses are available in **S1 File** and on the project GitHub repository (https://github.com/filiperomero2/CHIKV_RJ_2015-2018, last accessed on 30 May 2023).

## 2.4. Sample collection, RNA extraction, and CHIKV molecular diagnosis

To assess the genetic diversity of CHIKV in Rio de Janeiro state, we generated new CHIKV genomes obtained from samples collected in the Duque de Caxias municipality, the third most populous city in Rio de Janeiro state (**Fig 1A** and **1B**; https://cidades.ibge.gov.br/brasil/rj/duque-de-caxias/, last accessed 21 June 2022). A total of 179 samples, blood or urine, from arboviral suspected cases were collected in public health care units between July 2017 and June 2018. Viral nucleic acid extractions were performed with the QIAmp viral RNA mini kit (Qiagen, Germany), following manufacturer's instructions. We then tested extracted RNAs by RT-qPCR for detection of ZIKV, DENV, and CHIKV using the ZDC molecular kit (Bio-Manguinhos FIOCRUZ, Brazil). Samples presenting a cycle threshold (Ct) < 38 for a given target were considered positive for CHIKV.

## 2.5. Chikungunya virus whole-genome sequencing

We selected 34 CHIKV RNA-positive samples with Ct < 30 for viral whole-genome sequencing using the Illumina platform. Samples were distributed across the most sampled months of

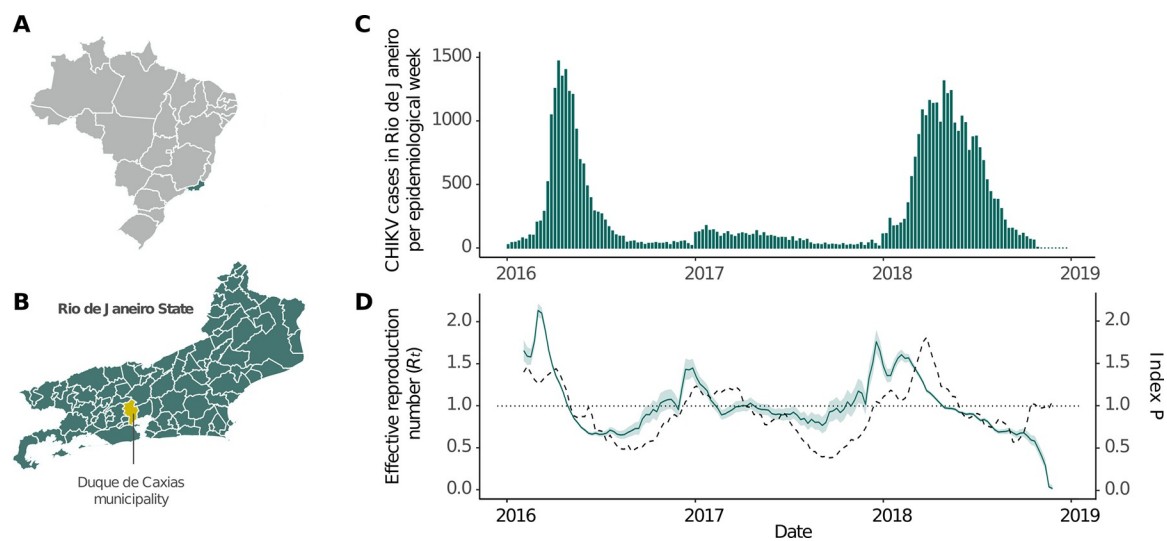

**Fig 1. Map of Brazil and Rio de Janeiro state, state-level weekly CHIKV incidence in the 2016–2018 period and $R_t$ estimate. (A)** Map of Brazil with all states colored in light gray, except for Rio de Janeiro, in green. **(B)** Rio de Janeiro map with municipality borders delimitation. The municipality of Duque de Caxias, where samples for genome sequencing were collected, is colored yellow. **(C)** State level weekly incidence data. Bars correspond to the number of CHIKV cases. **(D)** Estimate for $Rt$ between 2016 and 2018 (left y-axis). The solid line indicates mean values, while the ribbon indicates the 95% confidence interval. The dashed line represents index P values, a measure of transmissibility potential for the vector *Aedes aegypti* (right y-axis). The dotted line marks the critical epidemic threshold ($R_t$ = 1). Maps shapefiles were downloaded from Instituto Brasileiro de Geografia e Estatística (IBGE) at https://www.ibge.gov.br/en/geosciences/territorial-organization/territorial-meshes/18890-municipal-mesh.html (last accessed 26 June 2023).

2018 (March, April, and May), and one sample from 2017 was also sequenced. Amplicon target sequencing was conducted employing an amplicon-based sequencing protocol [39]. Briefly, we converted viral RNA to cDNA with SuperScript IV (Thermo Fisher Scientific), which was amplified with the Q5 DNA polymerase (New England Biolabs, UK) in two PCR reactions. Each PCR reaction contained 22 pairs of primers and generated overlapping amplicons of approximately 400 bp [39]. The amplicons covered the entire CHIKV genome and were used as input to the QIAseq 1-Step Library prep kit (Qiagen, Germany), following the manufacturer's protocol. Libraries were normalized and equimolarly pooled at 14 pMol with 10% phiX control. Sequencing was performed on an Illumina MiSeq instrument with a V3 (600 cycles) cartridge.

## 2.6. Virus genome assembly

Strict quality control of raw sequencing reads was performed with Trimmomatic v.0.39 [40], which removed low-quality bases (Phred scores < 30), sequencing adapters, and short reads (<50 nucleotides). This software was also used to conservatively trim the 30 initial nucleotides from all reads, eliminating primer-related sequences. Filtered reads from each sample were then mapped against a reference CHIKV genome (NCBI accession number: KP164568.1) with Bowtie2 v2.4.2 [41]. Mapping files were sorted and indexed with samtools v1.11 [42]. Bcftools v1.11 [43] was used to call variants with a high-quality threshold (QUAL>200) and to estimate consensus genome sequences. Finally, bedtools v2.30.0 [44] was used to mask low-coverage sites (<10x). This assembly pipeline is publicly available on GitHub (https://github.com/filiperomero2/ViralUnity, last accessed 20 June 2022). The novel consensus genome sequences belonged to the ECSA lineage and were deposited to NCBI GenBank under Accession Numbers OP312936 to OP312969 (**Table 1**).

**Table 1. Sample metadata and sequencing statistics for Duque de Caxias samples.**

| Sample ID | NCBI accession | Collection date | RT-qPCR Ct | Number of reads | Average depth | Genome coverage (%) |
|---|---|---|---|---|---|---|
| 10 | OP312936 | 2017-07-26 | 23.55 | 634,418 | 2582.12 | 85.84 |
| 53 | OP312937 | 2018-03-01 | 19.45 | 738,782 | 4041.52 | 97.24 |
| 57 | OP312938 | 2018-03-08 | 18.97 | 1,311,212 | 10039 | 98.00 |
| 66 | OP312939 | 2018-03-15 | 14.73 | 624,422 | 3281.5 | 94.34 |
| 71 | OP312940 | 2018-03-19 | 18.78 | 900,426 | 3051.64 | 70.50 |
| 79 | OP312941 | 2018-03-22 | 17.08 | 924,090 | 4257.63 | 95.15 |
| 83 | OP312942 | 2018-03-27 | 19.84 | 565,152 | 4016.5 | 96.19 |
| 89 | OP312943 | 2018-03-29 | 20.08 | 1,520,304 | 7710.58 | 94.39 |
| 92 | OP312944 | 2018-03-29 | 19.19 | 1,009,462 | 5340.61 | 97.68 |
| 97 | OP312945 | 2018-04-02 | 19.35 | 1,159,296 | 5811.29 | 96.74 |
| 99 | OP312946 | 2018-04-02 | 19.91 | 2,310,894 | 5509.86 | 84.85 |
| 101 | OP312947 | 2018-04-02 | 24.94 | 6,875,476 | 45036.2 | 95.21 |
| 107 | OP312948 | 2018-04-04 | 19.57 | 841,146 | 2703.55 | 84.18 |
| 110 | OP312949 | 2018-04-05 | 20.46 | 1,365,558 | 3543.14 | 78.29 |
| 114 | OP312950 | 2018-04-05 | 20.04 | 1,557,664 | 6901.02 | 93.02 |
| 116 | OP312951 | 2018-04-05 | 19.62 | 663,888 | 3471.67 | 93.19 |
| 117 | OP312952 | 2018-04-05 | 23.45 | 876,300 | 5417.87 | 90.16 |
| 120 | OP312953 | 2018-04-16 | 18.07 | 765,232 | 2285.81 | 94.73 |
| 126 | OP312954 | 2018-04-17 | 23.02 | 1,405,840 | 6905.76 | 87.42 |
| 128 | OP312955 | 2018-04-19 | 21.72 | 1,365,992 | 4532.91 | 84.24 |
| 130 | OP312956 | 2018-04-19 | 18.47 | 1,038,668 | 4913.46 | 94.99 |
| 134 | OP312957 | 2018-04-19 | 22.77 | 2,133,788 | 10443.2 | 94.98 |
| 143 | OP312958 | 2018-04-30 | 23.22 | 1,518,024 | 7322.5 | 91.43 |
| 149 | OP312959 | 2018-04-30 | 23.77 | 1,034,102 | 1806.96 | 85.10 |
| 152 | OP312960 | 2018-05-03 | 24.85 | 2,261,360 | 9137.28 | 87.06 |
| 155 | OP312961 | 2018-05-03 | 24.84 | 1,339,770 | 6698.01 | 85.11 |
| 157 | OP312962 | 2018-05-07 | 21.03 | 1,778,388 | 8387.42 | 89.23 |
| 159 | OP312963 | 2018-05-07 | 24.37 | 628,560 | 2628.08 | 90.56 |
| 161 | OP312964 | 2018-05-07 | 24.73 | 1,182,780 | 3850.23 | 89.45 |
| 165 | OP312965 | 2018-05-09 | 18.83 | 1,150,356 | 6482.64 | 96.64 |
| 166 | OP312966 | 2018-05-09 | 22.01 | 1,467,544 | 7224.4 | 86.22 |
| 170 | OP312967 | 2018-05-14 | 24.08 | 1,048,464 | 4201.88 | 91.86 |
| 174 | OP312968 | 2018-05-17 | 14.76 | 1,092,190 | 4108.37 | 95.44 |
| 181 | OP312969 | 2018-05-23 | 18.75 | 816,186 | 4507.98 | 97.82 |

## 2.7. Maximum likelihood phylogenetic analyses

To contextualize the novel genome sequences within the global CHIKV diversity, we downloaded all CHIKV sequences with more than 8,000 nucleotides available on NCBI GenBank [45], as of 31 May 2022. Sequences without information on the country of sampling and date of collection were removed from subsequent analyses. The remaining 1,262 sequences were combined with the 34 new sequences and aligned to the CHIKV RefSeq genome (NCBI accession: NC_004162.2) with MAFFT v7.480 [46]. After manually trimming gap-rich regions, we estimated maximum-likelihood (ML) phylogenetic trees using IQ-Tree v2.1.2 [47]. We used Model-Finder [48] to determine the best-fit substitution model and the Shimoidara-Hasegawa-like approximate likelihood ratio test (SH-aLRT) [49] to assess phylogenetic node support.

A second dataset was assembled, comprehending only sequences from the ECSA-American sub-lineage. Only Brazilian sequences collected up 2018 were kept in this dataset, as a preliminary analysis indicated international sequences or those collected later were not related to the genome sequences described in this study. Sequences were aligned against the oldest available ECSA strain from Brazil (Accession number: KP164568, from Feira de Santana, Bahia state), and non-coding regions were trimmed out of the alignment. A novel ML tree was inferred, using the methods described above. Screening of recombination signals was performed using RDP4 [50].

## 2.8. Temporal structure of CHIKV ECSA-American nucleotide alignment

Root-to-tip regressions were used to evaluate the temporal signal available on genomes from the ECSA-American sub-lineage dataset, including only Brazilian strains. The estimated maximum likelihood phylogenetic tree was manually rooted in the oldest available ECSA strain from Brazil and regression between divergence and sampling dates was analyzed in TempEst v1.5.3 [51]. To optimize the temporal signal, outliers were conservatively defined as sequences whose regression residuals exceeded more or less than two times the interquartile range of the residual distribution, and 51 sequences were removed. A novel ML tree inference and root-to-tip regression were performed on the filtered ECSA-American sub-lineage dataset ($n = 148$).

## 2.9. Bayesian phylogenetic analyses

To investigate the evolutionary origins of CHIKV in Rio de Janeiro, we used Bayesian coalescent and phylogeographic models available in BEAST v1.10.4 [52,53]. Analyses were run using a HKY+G4 nucleotide substitution model [54,55], a strict molecular clock model, and a flexible skygrid tree prior [56]. The cut-off for the skygrid model was set at 4.22 years, following a preliminary estimate of the time of the most recent common ancestor (tMRCA) obtained from TempEst (x-axis intercept in the regression). The number of grid points was set to match the number of months within the tree temporal span between the tMRCA and the earliest tip ($n = 51$). We also performed an alternative analysis with the uncorrelated relaxed clock model with lognormal rate distribution (UCLN) [57]. We used default priors and operators, and ran at least two independent Markov Chain Monte Carlo (MCMC) chains with 40 to 100 million steps, sampling every 10,000th step, using a maximum-likelihood phylogeny as a starting tree. The BEAGLE library was used for accelerated computations [58]. We assessed mixing of MCMC chains and convergence of all parameters using Tracer v1.7 [59]. We used Logcombiner [52] to sample 1,000 trees from the combined posterior trees distribution, after removal of 10% burn-in. A discrete phylogeographic analysis was then performed with a set of 1,000 empirical dated trees obtained from the posterior distribution, as previously described [60]. We used an asymmetric substitution model [53] and considered four geographic locations/regions: Rio de Janeiro state ($n = 67$), North region ($n = 22$), Northeast region ($n = 52$), and Central-West region ($n = 7$).

Finally, we also investigated changes in CHIKV genetic diversity over time in RJ. We assembled a third dataset, comprehending all Rio de Janeiro sequences belonging to a dominant clade, identified in the previous analysis ($n = 63$). We used a skygrid prior, as described above. In this case, we also used an informative prior on the root age based on the estimate obtained from the analysis of the full dataset. BEAST analyses were run in duplicate for 20 million generations, sampling every 2,000th state. BEAST XML files and outputs are available in **S2 File** and on the project GitHub repository (https://github.com/filiperomero2/CHIKV_RJ_2015-2018, last accessed on 30 May 2023).

## 2.10. Selection analyses and ancestral states reconstruction

Selection analyses were conducted on the ECSA-American alignment with Brazilian sequences described above using MEME (Mixed Effects Model of Evolution) [61] and FEL (fixed effects likelihood) [62] models to detect individual genome sites subjected to episodic and pervasive positive selection on the Datamonkey web server [63,64]. A significance threshold of $p < 0.01$ was used for analyses. To map positively selected sites along the evolutionary history of Brazilian CHIKV ECSA-American sub-lineage, we performed ancestral sequence reconstruction using TreeTime [65].

## 3. Results

### 3.1. State-level chikungunya incidence and transmissibility

The first chikungunya epidemic wave in Rio de Janeiro state occurred in 2016 (**Fig 1C**), with 16,245 laboratory-confirmed cases. In 2017, fewer cases were reported ($n$ = 3,989), with a nearly constant number of infections in the first semester, decreasing by the end of the year. In early 2018, a sharp increase in chikungunya infections were observed, leading to a larger epidemic with a magnitude greater than the first ($n$ = 24,990 cases). In both 2016 and 2018 waves, cases grew exponentially between January and March, reaching peaks in April. Most confirmed cases occurred in the state's capital, Rio de Janeiro city (54%; $n$ = 24,636/45,241; **S1 Fig**). This pattern was most evident in 2016 (87%; $n$ = 14,204/16,245) than in 2017 (44%; $n$ = 1,768/3,989) and 2018 (35%; $n$ = 8,649/24,990). In 2018, while the state capital exhibited the highest incidence, a large number of infections were reported in other municipalities, such as Campos dos Goytacazes (24%; $n$ = 5,968/24,990) and São Gonçalo (21%, $n$ = 5,323/24,990).

To better understand the transmission dynamics of CHIKV ECSA in Rio de Janeiro state, we estimated $R_0$ from notification data. First, we estimated an epidemic growth rate, $r$, of 1.032 (95% CI: 1.027–1.038) for 2016. Assuming a generation time (GT) of 14 days, we estimated a $R_0$ of 1.56 (95% CI = 1.46–1.67). Sensitivity analysis with different GT distributions generated different $R_0$ values (GT mean = 10 days, $R_0$ = 1.37, 95% CI: 1.31–1.44; GT mean = 20 days, $R_0$ = 1.89, 95% CI: 1.72–2.09) (**S2 Fig**). Overall, these estimates are similar to those previously obtained for DENV and lower compared to $R_0$ estimates obtained for the ZIKV epidemic in Rio de Janeiro [66].

We then estimated the $Rt$ to track the progress of the epidemic in Rio de Janeiro state (**Fig 1D**). Noticeably, we observed that between 2016 and mid-2018, high transmissibility periods broadly coincide with periods of relatively high *Ae. aegypti* transmission potential. This shows that during the study period, the timing of chikungunya recurrence in Rio de Janeiro was at least partly driven by climatic factors such as temperature and humidity that correlate with fluctuations in *Ae. aegypti* abundance (**Figs 1D and S3**). As expected, $R_t$ follows a seasonal pattern, reaching the peak values in early 2016 (mean: 2.14, 95% CI: 2.06–2.22), 2017 (mean: 1.45, 95% CI: 1.35–1.56) and 2018 (mean: 1.76, 95% CI: 1.63–1.91). $R_t$ estimates obtained using different window lengths, and GT distributions revealed similar patterns (**S4 Fig**).

### 3.2. Molecular detection and whole-genome sequencing of CHIKV from Duque de Caxias municipality

Of the 179 clinical samples from patients presenting symptoms compatible with arbovirus infections in health-care units of the Duque de Caxias municipality, 48.6% ($n$ = 87) tested positive for CHIKV RNA (mean cycle threshold, Ct, was 23.2, range: 14.7 to 33.1), 1.1% ($n$ = 2) tested positive for DENV and 2.8% ($n$ = 5) for ZIKV. We detected CHIKV coinfections with DENV ($n$ = 1) and ZIKV ($n$ = 3). For the CHIKV RNA-positive cases, clinical manifestations

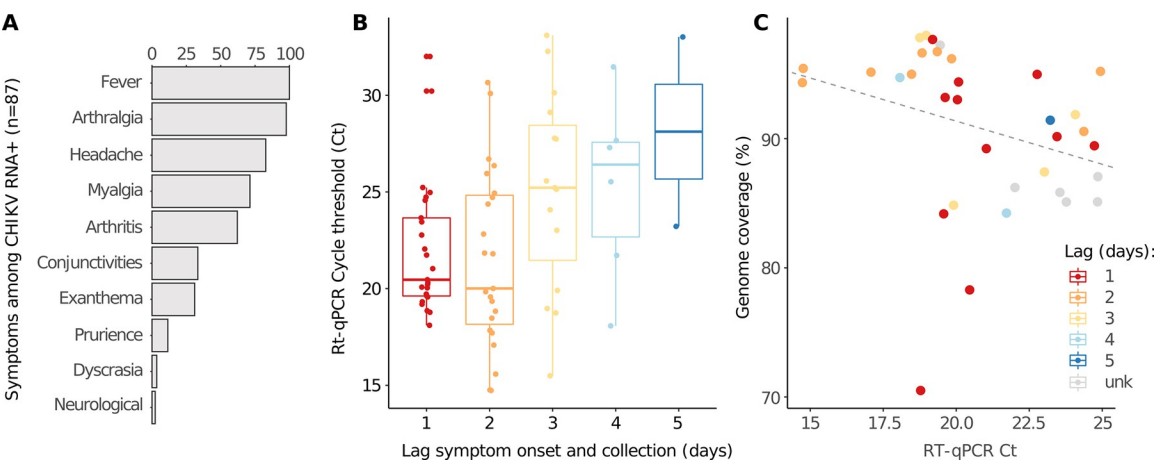

**Fig 2. Symptoms distribution, RT-qPCR Ct values dynamics and their correlation with genome sequencing coverage. (A)** Distribution of symptoms exhibited by all CHIKV positive patients in the Duque de Caxias cohort. **(B)** The time lag between symptoms onset and sample collection dates exhibits correlation with RT-qPCR Ct values. As infection proceeds, viral loads decrease (Cts increase) likely due to immunological response. **(C)** Negative correlation between RT-qPCR Cts and genome sequencing coverage. Sequences characterized from samples with higher viral load (lower Cts) tend to exhibit higher coverage, although no strong statistical correlation was inferred on a linear model ($p = 0.08$).

included fever (100%, $n = 87/87$), arthralgia (97.7%, $n = 85/87$), headache (82.8%, $n = 72/87$), myalgia (71.3%, $n = 62/87$), arthritis (62.1%, n = 54/87), conjunctivitis (33.3%, $n = 29/87$), and exanthema (31%, $n = 27/87$) (**Fig 2A** and **S3 File**). Less frequently, prurience (11.5%, $n = 10/87$), dyscrasia (3.4%, $n = 3/87$) and neurological symptoms (2.3%, $n = 2/87$) were also recorded. As expected, we observed a tendency towards lower values (higher viral loads) in samples collected closer to the dates of symptom onset (**Fig 2B**). Thirty-four near complete CHIKV genome sequences were generated with an average 90.8% (standard deviation: 6.2%, range: 70.5–98%) genome coverage in relation to the CHIKV genome reference sequence (**Table 1**). Sequencing coverage was inversely correlated with Ct values, although no strong statistical relationship was inferred on a linear model ($p = 0.08$; **Fig 2C**).

### 3.3. Phylogenetic analyses of the CHIKV ECSA-American lineage

Newly generated sequences clustered within the ECSA-American lineage with maximum statistical support (SH-aLRT = 100; **Fig 3A**). After removal of outliers, we found a strong correlation between sampling dates and genetic divergence (**Fig 3B and 3C**; $R^2 = 0.72$). No recombination was detected. Most of the new sequences ($n = 28/31$) clustered within a dominant clade of sequences from Rio de Janeiro state (named RJ1; $n = 74$, SH-aLRT = 88.4), while a few ($n = 3/31$) grouped in a smaller clade (RJ2; $n = 4$, SH-aLRT = 95.2) together with a publicly available sequence from Rio de Janeiro. Most sequences from Duque de Caxias clustered with sequences from other municipalities, such as Niterói and São João de Meriti, suggesting frequent viral spread within state borders.

Time scale inferences with both strict and UCLN models results were highly congruent. As the posterior distribution of the coefficient variation of the UCLN model displayed wide variation and included extremely low values (95% BCI: $6.68 \times 10^{-4}$–0.43), we opted for focusing on the strict model (**S2 File**). Our time-scaled phylogenetic analysis estimated an evolutionary rate of around $5.72 \times 10^{-4}$ (95% Bayesian credible interval, BCI, $4.89 \times 10^{-4}$–$6.50 \times 10^{-4}$) substitutions per site per year and corroborated that the ECSA-American clade emerged in the Northeast region in mid-2014 (mean: 21 July, 95% BCI: 29 May to 25 August; **Fig 4A**).

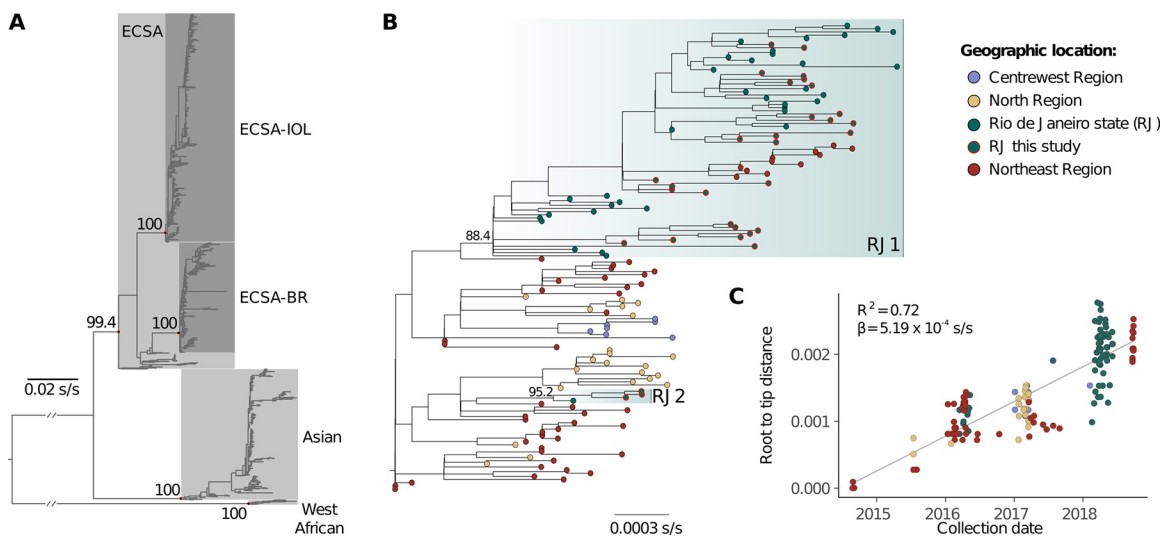

**Fig 3. Maximum likelihood phylogenetic analysis of global and ECSA-American datasets. (A)** Phylogenetic tree inferred from the global dataset. All new characterized genomes clustered within the ECSA-Br clade. Names of lineages and relevant clades are indicated. SH-aLRT statistical support values for these clades are shown close to their defining nodes (colored in red). **(B)** Phylogeny inferred from the filtered ECSA-American dataset. Tip shapes are colored according to sampling location (Centre-West region: purple, North region: yellow, Northeast region: red, Rio de Janeiro state: green). Sequences generated in this study are highlighted with red circles around the tip shapes. Clades composed mostly by RJ sequences are indicated along their SH-aLRT support values. **(C)** The root-to-tip regression plot, which indicates a strong temporal signal ($R^2 = 0.72$, slope = $5.19 \times 10^{-4}$). Scale bars represent substitutions per site (s/s).

CHIKV-ECSA-American clade spread to the North and Centre-West regions of Brazil, as well as to Rio de Janeiro state. RJ1 and RJ2 clades result from independent introductions from the Northeast region with location posterior probability support > 99%. In contrast, the tMRCA for RJ1 was estimated in mid-2015 (mean: 7 July, 95% BCI: 4 April to 24 September 2015; clade posterior probability: 0.99), RJ2 tMRCA was dated around mid-2017 (mean: 7 July 2017, 95% BCI: 12 March 2017 to 6 November 2017; clade posterior probability: 1.00). Although we detected a more recent introduction from the Northeast region, our data suggests that the 2018 chikungunya epidemic in Rio de Janeiro was driven mostly by the RJ1 clade that was circulating in Rio de Janeiro since mid-2015.

The demographic pattern of the RJ1 clade inferred through coalescent non-parametric analysis, revealed that virus genetic diversity rose during the 2016 epidemic, declined through 2017, and rose again (**Fig 4B**). Notably, epidemiological and genetic data independently suggest that 2017 had a much lower number of infections than 2016 and 2018, despite the circulation of locally established lineage (RJ1) and a window of climatic suitability for transmission (**Fig 1D**).

## 3.4. Adaptive molecular evolution of CHIKV ECSA-American

Our ancestral reconstruction revealed 93 mutations associated with RJ lineages, of which 75 were synonymous and 18 non-synonymous (**S5 Fig**). Non-synonymous mutations have been identified in nearly all CHIKV ORFs: nsP1 (R76K, D531G), nsP2 (P16L, G261R, N299S, A545S, V645A, T652I), nsP3 (T422I, R435H), nsP4 (R99Q, I111V, S157C), C (K89R, A264T), E2 (V408I) and E1 (A305T, V669M). MEME and FEL models detected three and two amino acid changes under positive selection with statistical significance ($p < 0.01$; see also **S4 File**), respectively. Both models indicate that amino acids nsP1: D531G and nsP4: A481D were under positive selection. Interestingly, nsP4: A481D is a lineage-defining mutation of a clade

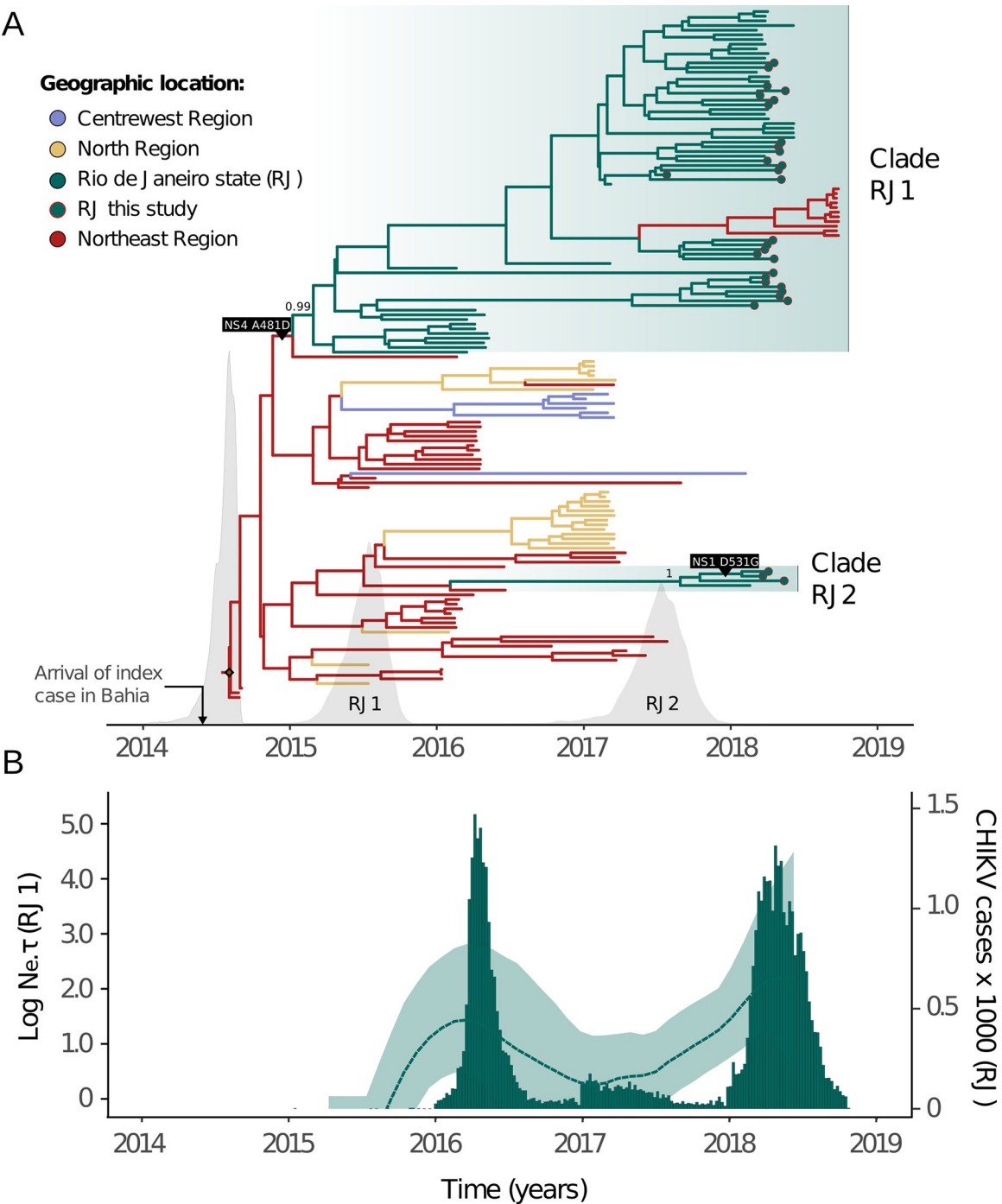

**Fig 4. Bayesian time scaled phylogeographic reconstruction for the clade ECSA-Br. (A)** The molecular clock phylogeny annotated with discrete trait reconstructions. Colors indicate estimated ancestral locations (Centre-West region: purple, North region: yellow, Northeast region: red, Rio de Janeiro state: green). Tip shapes mark sequences generated in this study. The x-axis depicts the timescale, while the density plots indicate the posterior distributions estimated for the age of clades ECSA-American, RJ1 and RJ2. Posterior probabilities for both RJ clades are shown. Positively selected mutations detected with MEME and FEL models (NS1: D351G and NS4: A481D) are exhibited on the branches where they occurred according to the ancestral states reconstruction performed (TreeTime). The inset marks the date of arrival of the index case in the Northeast region. **(B)** Skygrid reconstruction plot for the clade RJ1. A separate analysis was performed with only RJ sequences from the clade RJ1, allowing the reconstruction of the dynamics of variation of viral effective population size in the state (left y-axis). The analysis reveals CHIKV genetic diversity varied over time, with periods of high diversity matching peaks observed in incidence data (right y-axis).

that comprehends lineage RJ1 and a single sequence from Sergipe state (Northeast region, Accession Number: KY055011) (**Figs 4A and S5A**). In contrast, the nsP1: D531G amino acid change occurs within a subclade of RJ2, but also emerged multiple independent times within RJ1 (**Figs 4A and S5A**).

## 4. Discussion

We provide a detailed picture of the epidemic history of CHIKV ECSA-American in Rio de Janeiro state from 2016 to 2018, which could be divided into three epidemic phases. First, our analyses suggest that CHIKV ECSA was introduced in Rio de Janeiro around mid-2015, causing its first recorded epidemic in 2016, once ZIKV cases started decreasing in the state [67]. We estimated a $R_0$ for CHIKV ECSA-American in Rio de Janeiro to be around 1.56, similar to those reported to CHIKV across other settings [18,68–70]. Additionally, our $R_0$ for CHIKV was consistent with estimates for DENV in Rio de Janeiro [66], and lower compared to estimates for Zika [66], which caused large epidemics in Rio de Janeiro between 2015–2016 [71]. However, we then found unusually low CHIKV incidence and genetic diversity in 2017. The implementation of large-scale *Wolbachia* interventions in *Ae. aegypti* populations in Rio de Janeiro only started in late 2017 in a single neighborhood (*Ilha do Governador*); therefore, it is unlikely to explain the observed patterns [72]. Low chikungunya incidences in 2017 were also observed in a large serosurveillance study of blood donors [73]. Thus, we hypothesize that the low number of cases in 2017 can partially be explained by climatic factors since *Ae. aegypti* suitability, mean temperature, or humidity in 2017 was lower compared to previous and following years. Moreover, the large chikungunya epidemic in Rio de Janeiro in 2018 indicates that the trend observed in 2017 was unlikely to be caused primarily by population immunity. Indeed, surveys found that 18% of seroprevalence against CHIKV in >2,000 participants screened between July and October 2018 [74]. Additional investigations into the dynamics of arbovirus population immunity, including accurate estimates of force of infection, and analysis of data on vector control policies and population behavior, could help better understand the dynamics and evolution of CHIKV transmission in the Americas.

Overall, our results show that CHIKV incidence, $R_t$, and *Ae. aegypti* transmission potential display a seasonal pattern and are generally synchronized. Similar trends have been noticed for ZIKV [18,26], and further highlight the interplay between vector ecology, climate and arbovirus disease transmission. Given its major connectivity at the national and international level, Rio de Janeiro is a key hotspot for arbovirus transmission and a location of interest for epidemiological surveillance in Brazil. For instance, several events of dengue lineage replacement were caused by novel lineages initially detected in Rio de Janeiro [27]. In 2018, three different arboviruses circulated in Rio de Janeiro, CHIKV, DENV, and ZIKV, with CHIKV causing the most infections. We noticed a similar scenario in the Duque de Caxias municipality, from which 48.6% were positive for CHIKV and much smaller proportions for ZIKV (2.79%) and DENV (1.11%) in 2018. Overall, clinical symptoms in our cohort were similar to those described elsewhere [75]. However, among the CHIKV RNA-positive cases, we recorded one presenting neurological manifestations. A recent study has shown that neurological signs and symptoms were reported in 39% of CHIKV cases with a fatal outcome [3]. Additional studies are needed to investigate host and viral factors associated with clinical manifestations.

Our genetic analyses confirm that the clade ECSA-American lineage emerged in Northeast Brazil in mid-2014, consistent with the arrival of the index case in Bahia state in late May [16,17]. This lineage then spread to all other regions over the following couple of years, being introduced in Rio de Janeiro around mid-2015, and later in mid-2017. We show that both introductions occurred from the Northeast region, consistent with previous findings

[20,21,29]. As the first autochthonous cases of CHIKV in Rio de Janeiro were detected in November 2015, the period of cryptic transmission in the state was between two and seven months, consistent with previous estimates [20,29].

Our analyses show the acquisition of a range of non-synonymous mutations in both non-structural and structural genes of the ECSA-American lineage. Although *Ae. albopictus* mosquitoes are abundant in Rio de Janeiro [76], none of the new sequences presented the mutations E1:A226V or E2:L210Q, previously associated with increased transmission competence by *Ae. albopictus* mosquitoes in the Indian Ocean Lineage [11,12]. However, it remains unclear whether other mutations in CHIKV ECSA-American strains are associated with increased transmissibility in *Ae. aegypti* and/or *Ae. albopictus*. Notwithstanding, our analyses prove that at least two non-synonymous mutations associated with sequences from Brazil evolved through molecular adaptation: nsP1: D351G and nsP4: A481D. While evidence for positive selection in non-structural genes has already been identified for the CHIKV Asian genotype [77], this is the first report of this phenomenon for CHIKV ECSA. Signals of positive selection in non-structural proteins have also been detected for another alphavirus (Western equine encephalitis virus) in natural settings [78,79], and experiments with chimeric alphaviruses have shown that adaptive mutations in nonstructural proteins are essential for viral replication and infectivity [80]. Nonstructural proteins have also been shown to be preferential targets for adaptive molecular evolution among different flaviviruses [81]. As these new mutations may impact the virus' biological properties and epidemiology, further studies with cellular and animal models, including *Ae. aegypti* and *Ae. albopictus* vector competence surveys, should be performed to clarify their phenotypic impact.

In conclusion, our study sheds light on the epidemiological dynamics of chikungunya in the state of Rio de Janeiro up to 2018. We demonstrate that chikungunya incidence and transmissibility varied seasonally, following trends in *Ae. aegypti* transmission potential. We also provided first reports of CHIKV ECSA-American epidemiological parameters and identified mutations of interest that evolved under positive selection. We encourage scaling up diagnostic efforts for arboviruses and deployment of effective mosquito control strategies. Larger forthcoming genomic and epidemiological CHIKV-ECSA studies will help to disentangle the impact of virus molecular evolution and adaptation, host population behavior and immunity, and vector ecology to improve public health interventions against arboviruses.

## Supporting information

**S1 Fig. CHIKV incidence within Rio de Janeiro state. (A)** CHIKV incidence per state region. Rio de Janeiro is officially divided into five intermediate regions, each comprehending between 12 and 26 municipalities (IBGE, https://www.ibge.gov.br/apps/regioes_geograficas/, last accessed 17 August 2022). **(B)** CHIKV incidence per municipality. As the state comprehends 92 municipalities, and many of them present low incidence levels through the entire period, individual color legends have been omitted. Municipalities that at any point presented weekly incidence above 500 cases were highlighted in the plot (Campos dos Goytacazes, Rio de Janeiro, São Gonçalo).
(TIFF)

**S2 Fig. $R_0$ estimates and sensitivity analysis.** $R_0$ estimates obtained with epidemiological data under different generation time distributions (gamma distributions with means 10, 14 and 20, and constant standard deviation: 6.4 days).
(TIFF)

**S3 Fig.** Time series of climatic variables—**(A)** 2m temperature, **(B)** total precipitation and **(C)** relative humidity—for Rio de Janeiro state between 2014 and 2018. All climatic data was obtained from the ECMWF reanalysis product [34].
(TIFF)

**S4 Fig. $R_t$ estimates and sensitivity analysis. (A)** $R_t$ estimate performed with different generation time (GT) distributions (gamma distributions with means 10, 14 and 20, and constant standard deviation: 6.4 days). Colors indicate estimates for different GTs (10: gray, 14: yellow, 20: blue). **(B)** $R_t$ estimates performed with different sliding window lengths (3 to 8 weeks). Colors indicate estimates for different window lengths (3: salmon, 4: yellow, 5: green, 6: blue, 7: purple, 8: pink). Solid lines indicate mean values and ribbons indicate the 95% confidence intervals. The dashed lines denote the critical epidemic threshold ($R_t = 1$).
(TIFF)

**S5 Fig. TreeTime and HyPhy analysis. (A)** The molecular clock tree had its branches colored according to the ancestral location's reconstructions. Tip shapes indicate sequences generated in this study. Light gray ticks along branches indicate synonymous mutations, while dark gray ticks mark non-synonymous mutations. Mutations for which evidence of positive selection was found (MEME and FEL models) are marked in red (NS4: A481D) and blue (NS1: D531G). **(B)** CHIKV genome scheme exhibiting all mutations reconstructed and associated with RJ clades. Non-synonymous mutations are numbered from 1 to 18 and their positions and corresponding amino acid replacements are indicated by black vertical lines. Similarly, vertical gray lines indicate the position of the synonymous mutations inferred. Non-structural and structural open reading frames are colored in dark green and light green, respectively.
(TIFF)

**S1 File. Data and R code used in epidemiological analyses.**
(ZIP)

**S2 File. BEAST xml files and outputs generated in this study.**
(ZIP)

**S3 File. Clinical data from CHIKV positive patients.**
(XLSX)

**S4 File. Input and output files of selection analysis with Datamonkey.**
(ZIP)

## Acknowledgments

We thank the Health Department of the Municipality of Duque de Caxias, RJ, Brazil, which allowed the collection of clinical samples from patients with suspected arboviruses.

## Author Contributions

**Conceptualization:** Marcelo Calado de Paula Tôrres, Amilcar Tanuri, Sergian Vianna Cardozo, Nuno Rodrigues Faria, Renato Santana Aguiar.

**Data curation:** Filipe Romero Rebello Moreira, Mariane Talon de Menezes, Clarisse Salgado-Benvindo, Hury Hellen Souza de Paula, André Frederico Martins, Raphael Rangel das Chagas, Rodrigo Decembrino Vargas Brasil, Felipe Iani, Sergian Vianna Cardozo.

**Formal analysis:** Filipe Romero Rebello Moreira, Charles Whittaker, Victoria Cox, Nilani Chandradeva, Darlan da Silva Cândido, Ilaria Dorigatti, Oliver Brady, Carolina Moreira Voloch, Nuno Rodrigues Faria.

**Funding acquisition:** Amilcar Tanuri, Renato Santana Aguiar.

**Investigation:** Filipe Romero Rebello Moreira, Mariane Talon de Menezes, Clarisse Salgado-Benvindo, Charles Whittaker, Victoria Cox, Nilani Chandradeva, Hury Hellen Souza de Paula, André Frederico Martins, Raphael Rangel das Chagas, Rodrigo Decembrino Vargas Brasil, Alice Laschuk Herlinger, Patricia Alvarez, Marcelo Calado de Paula Tôrres, Ilaria Dorigatti, Oliver Brady, William Marciel de Souza, Sergian Vianna Cardozo.

**Methodology:** Filipe Romero Rebello Moreira, Mariane Talon de Menezes, Clarisse Salgado-Benvindo, Darlan da Silva Cândido, Marisa de Oliveira Ribeiro, Monica Barcellos Arruda, Carolina Moreira Voloch, Sergian Vianna Cardozo, Nuno Rodrigues Faria, Renato Santana Aguiar.

**Project administration:** Sergian Vianna Cardozo, Renato Santana Aguiar.

**Resources:** Carolina Moreira Voloch, Amilcar Tanuri, Nuno Rodrigues Faria, Renato Santana Aguiar.

**Software:** Filipe Romero Rebello Moreira, Darlan da Silva Cândido.

**Supervision:** Sergian Vianna Cardozo, Nuno Rodrigues Faria, Renato Santana Aguiar.

**Visualization:** Filipe Romero Rebello Moreira, Nuno Rodrigues Faria.

**Writing – original draft:** Filipe Romero Rebello Moreira, Nuno Rodrigues Faria.

**Writing – review & editing:** Filipe Romero Rebello Moreira, Mariane Talon de Menezes, Clarisse Salgado-Benvindo, Charles Whittaker, Victoria Cox, Nilani Chandradeva, Hury Hellen Souza de Paula, André Frederico Martins, Raphael Rangel das Chagas, Rodrigo Decembrino Vargas Brasil, Darlan da Silva Cândido, Alice Laschuk Herlinger, Marisa de Oliveira Ribeiro, Monica Barcellos Arruda, Patricia Alvarez, Marcelo Calado de Paula Tôrres, Ilaria Dorigatti, Oliver Brady, Carolina Moreira Voloch, Amilcar Tanuri, Felipe Iani, William Marciel de Souza, Sergian Vianna Cardozo, Nuno Rodrigues Faria, Renato Santana Aguiar.

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
