## [Decision Letter · Decision Letter 0]

25 May 2023

Dear Mr. Moreira,

Thank you very much for submitting your manuscript "Epidemiological and genomic investigation of chikungunya virus in Rio de Janeiro state, Brazil, between 2015 and 2018" for consideration at PLOS Neglected Tropical Diseases. As with all papers reviewed by the journal, your manuscript was reviewed by members of the editorial board and by several independent reviewers. The reviewers appreciated the attention to an important topic. Based on the reviews, we are likely to accept this manuscript for publication, providing that you modify the manuscript according to the review recommendations. 

According to the reviewers, the current research is valuable and well-conducted. In the revised version, clarifications of novelty of the current research when compared to other similar studies in the same geographic region, the validity of the selected priors for the phylogenetic analyses, and the availability of various dataset should be made.

Sincerely,

Qu Cheng, Ph.D.

Guest Editor

Elvina Viennet

Section Editor

According to the reviewers, the current research is valuable and well-conducted. They also commented on the novelty of the current research when compared to other similar studies in the same geographic region, the validity of the selected priors for the phylogenetic analyses, the availability of various dataset, etc.

Reviewer's Responses to Questions

**Key Review Criteria Required for Acceptance?**

**Methods**

-Are the objectives of the study clearly articulated with a clear testable hypothesis stated?

-Is the study design appropriate to address the stated objectives?

-Is the population clearly described and appropriate for the hypothesis being tested?

-Is the sample size sufficient to ensure adequate power to address the hypothesis being tested?

-Were correct statistical analysis used to support conclusions?

-Are there concerns about ethical or regulatory requirements being met?

Reviewer #1: (No Response)

Reviewer #2: Minor Revision

Reviewer #3: A detailed description of the process to support the research findings

**Results**

-Does the analysis presented match the analysis plan?

-Are the results clearly and completely presented?

-Are the figures (Tables, Images) of sufficient quality for clarity?

Reviewer #1: (No Response)

Reviewer #2: Major Revision

Reviewer #3: Epidemiology and modeling – recommend to update information (if available)

• the case-fatality ratio?

• the case-hospitalization ratio?

**Conclusions**

-Are the conclusions supported by the data presented?

-Are the limitations of analysis clearly described?

-Do the authors discuss how these data can be helpful to advance our understanding of the topic under study?

-Is public health relevance addressed?

Reviewer #1: (No Response)

Reviewer #2: Major Revision

Reviewer #3: Recommend to strengthen discussion on spatial analysis:

• Not all hotspots are equal, as brought up in a recent paper on a dengue-related risk study that locations of mass public transit such as train stations have known to be hotpots for dengue transmission compared to homes in urban regions (Lefebvre et al., ijerph 2022)

• Comparisons between neighborhoods: are there any data on differences in R0 for different regions of Rio de Janeiro e.g. comparing transmissibility of CHIKV in favelas versus suburbia

**Editorial and Data Presentation Modifications?**

Reviewer #1: Minor Revision

Reviewer #2: Minor Revision

Reviewer #3: Accept

**Summary and General Comments**

Reviewer #1: Moreira et al. investigated the transmission dynamics of the chikungunya virus in Rio de Janeiro state, Brazil, between 2015 and 2018 in this study.

Minor comments:

Could the authors provide an explanation for why they chose the strict molecular clock model and a flexible skygrid tree prior for the Bayesian phylogenetic analyses?

Could the authors include more information about the various datasets they constructed for each analysis?

Although the authors have provided input and R codes for the epi-analysis, I would like to see all input, metadata, and output files for each generated figure (F1-4 and Fig S1-4) publicly accessible in a repository such as GitHub before acceptance. I hope the authors recognize that this is essential for open-access science and reproducibility.

Reviewer #2: Moreira FRR et al. revealed the epidemic and evolutionary history of Chikungunya virus in Rio de Janeiro state from 2016 to 2018. This is a valuable approach, and this manuscript uses a dataset that has a lot of potential for future work. I have several concerns and comments about the present analysis.

1. Previous studies have documented the epidemiological and evolution characteristics of Chikungunya virus in the city of Rio de Janeiro. Therefore, the author needs to highlight the innovative points worth publishing in this article.

(1) Souza, T.M.L., Vieira, Y.R., Delatorre, E. et al. Emergence of the East-Central-South-African genotype of Chikungunya virus in Brazil and the city of Rio de Janeiro may have occurred years before surveillance detection. Sci Rep 9, 2760 (2019). https://doi.org/10.1038/s41598-019-39406-9

(2) Xavier J, Giovanetti M, Fonseca V, Thézé J, Gräf T, Fabri A, Goes de Jesus J, Lima de Mendonça MC, Damasceno Dos Santos Rodrigues C, Mares-Guia MA, Cardoso Dos Santos C, Fraga de Oliveira Tosta S, Candido D, Ribeiro Nogueira RM, Luiz de Abreu A, Kleber Oliveira W, Campelo de Albuquerque CF, Chieppe A, de Oliveira T, Brasil P, Calvet G, Carvalho Sequeira P, Rodrigues Faria N, Bispo de Filippis AM, Alcantara LCJ. Circulation of chikungunya virus East/Central/South African lineage in Rio de Janeiro, Brazil. PLoS One. 2019 Jun 11;14(6):e0217871.

2. More information would be obtained if the dynamic pattern of the Chikungunya virus could be further explored based on the kinetic mechanism model, which integrated the epidemiological data, environmental data and molecular data in one model. The authors should discuss this topic in depth. 

Henderson AD, Kama M, Aubry M, Hue S, Teissier A, Naivalu T, Bechu VD, Kailawadoko J, Rabukawaqa I, Sahukhan A, Hibberd ML, Nilles EJ, Funk S, Whitworth J, Watson CH, Lau CL, Edmunds WJ, Cao-Lormeau VM, Kucharski AJ. Interactions between timing and transmissibility explain diverse flavivirus dynamics in Fiji. Nat Commun. 2021 Mar 15;12(1):1671.

3. The authors analyze the possible causes of low prevalence of Chikungunya virus in 2017 compared with 2016 and 2018, in the discussion section, and concluded that the trend observed in 2017 was unlikely to be caused by population immunity. Rather, it is more likely to be caused by environmental factors. Theoretically, the 2016 pandemic would have led to a reduction in the susceptible population, which in turn reduced the intensity of the pandemic in 2017. It is very important to better distinguish the role of each driver.

4. The author hypothesized that the low number of cases in 2017 can partially be explained by climatic factors since Ae. aegypti suitability, mean temperature, or humidity in 2017 was lower compared to previous and following years (page 22). The authors only presented the index P in Fig.1, however, there was no detailed description of the index P and its relation to Rt value in the Result part. Besides, the author mentioned the potential drivers may be mean temperature, or humidity. Can the author place the time-series of mean temperature, humidity and precipitation in the supplementary material?

5. The line number is missing in the manuscript. 

6. The results regarding to the clinical presentation and the genome sequencing appear discordant when placed together.

Reviewer #3: The authors did a great job in analyzing the impact of CHIKV during an outbreak in Rio de Janeiro using scalable data such as R(o) and R(t). The authors have brought up under introduction that urbanization is a contributor to CHIKV transmission and its diversity. Adding additional statements (or analysis) under discussion such as spatial analysis stratified by regions within Rio would enrich the conclusions, providing insight into the socioeconomic determinants that may contribute to the overall transmissibility of CHIKV-ECSA in a hyper-urbanized city such as Rio de Janeiro.

Introduction - “Chikungunya’s geographic distribution – particularly of Asian, IOL and ECSA lineages has been expanding rapidly over the last 20 years probably due rapid urbanization, globalization of trade, and virus evolution and adaptation to local variation in the distribution of vector species”

• To check for a more fitting reference for this statement

• “Due to air and fluvial connectivity and high mosquito climatic suitability” - To clarify on the suitable climatic conditions in Rio de Janeiro

PLOS authors have the option to publish the peer review history of their article (what does this mean?). If published, this will include your full peer review and any attached files.

Reviewer #1: No

Reviewer #2: No

Reviewer #3: No

Figure Files:

Data Requirements:

Reproducibility:

References

---

## [Decision Letter · Decision Letter 1]

17 Jul 2023

Dear Mr. Moreira,

We are pleased to inform you that your manuscript 'Epidemiological and genomic investigation of chikungunya virus in Rio de Janeiro state, Brazil, between 2015 and 2018' has been provisionally accepted for publication in PLOS Neglected Tropical Diseases.

Best regards,

Qu Cheng, Ph.D.

Guest Editor

Elvina Viennet

Section Editor

Reviewer's Responses to Questions

**Key Review Criteria Required for Acceptance?**

**Methods**

-Are the objectives of the study clearly articulated with a clear testable hypothesis stated?

-Is the study design appropriate to address the stated objectives?

-Is the population clearly described and appropriate for the hypothesis being tested?

-Is the sample size sufficient to ensure adequate power to address the hypothesis being tested?

-Were correct statistical analysis used to support conclusions?

-Are there concerns about ethical or regulatory requirements being met?

Reviewer #1: The authors have thoroughly addressed all the comments raised during the initial round of revision. The resulting manuscript appears to be exceptionally well-designed and is deemed suitable for publication.

Reviewer #2: Yes

Reviewer #3: (No Response)

**Results**

-Does the analysis presented match the analysis plan?

-Are the results clearly and completely presented?

-Are the figures (Tables, Images) of sufficient quality for clarity?

Reviewer #1: The authors have thoroughly addressed all the comments raised during the initial round of revision. The resulting manuscript appears to be exceptionally well-designed and is deemed suitable for publication.

Reviewer #2: Yes

Reviewer #3: (No Response)

**Conclusions**

-Are the conclusions supported by the data presented?

-Are the limitations of analysis clearly described?

-Do the authors discuss how these data can be helpful to advance our understanding of the topic under study?

-Is public health relevance addressed?

Reviewer #1: The authors have thoroughly addressed all the comments raised during the initial round of revision. The resulting manuscript appears to be exceptionally well-designed and is deemed suitable for publication.

Reviewer #2: Yes

Reviewer #3: (No Response)

**Editorial and Data Presentation Modifications?**

Reviewer #1: The authors have thoroughly addressed all the comments raised during the initial round of revision. The resulting manuscript appears to be exceptionally well-designed and is deemed suitable for publication.

Reviewer #2: Accept

Reviewer #3: (No Response)

**Summary and General Comments**

Reviewer #1: The authors have thoroughly addressed all the comments raised during the initial round of revision. The resulting manuscript appears to be exceptionally well-designed and is deemed suitable for publication.

Reviewer #2: no comments

Reviewer #3: Based on their revisions, I believe that authors have aptly revised their work to reflect our suggested comments.

PLOS authors have the option to publish the peer review history of their article (what does this mean?). If published, this will include your full peer review and any attached files.

Reviewer #1: No

Reviewer #2: No

Reviewer #3: No

---

## [Editor Report · Acceptance letter]

25 Sep 2023

Dear Mr. Moreira,

We are delighted to inform you that your manuscript, "Epidemiological and genomic investigation of chikungunya virus in Rio de Janeiro state, Brazil, between 2015 and 2018," has been formally accepted for publication in PLOS Neglected Tropical Diseases.

Best regards,

Shaden Kamhawi

co-Editor-in-Chief

Paul Brindley

co-Editor-in-Chief
